# Video Q-Former: Multimodal Large Language Model with Spatio-Temporal Querying Transformer Towards Video Understanding

## Abstract

Large language models (LLMs) have made remarkable strides in natural language processing tasks. However, effectively processing and understanding visual information remains a challenge for these models. To address this, multimodal large language models have been proposed, which integrate pre-trained visual encoders with LLMs. Although existing image-based approaches have shown success in aligning visual and textual modalities, extending these advancements to videos is challenging due to the richer visual and temporal information they contain. Current methods, including Video-ChatGPT and Video-LLaMA, have limitations in capturing inter-frame relationships and providing sufficient semantic context. To overcome these challenges, we propose Video Q-Former, a model that adaptively extracts spatiotemporal features from videos with a spatio-temporal querying transformer, enhancing the LLM's comprehension of visual-language alignment. Extensive experiments demonstrate that our model achieves state-of-the-art performance across various datasets in zero-shot video question answering tasks.

## 1 Introduction

In recent years, large language models (LLMs) Touvron et al. (2023a;b); Chowdhery et al. (2022); OpenAI (2023); Bai et al. (2022) have achieved remarkable success. These models have undergone extensive unsupervised pre-training on a substantial amount of unlabeled text data and have been supplemented with supervised fine-tuning and reinforcement learning from human feedback (RLHF) methods. As a result, they have developed exceptional text comprehension capabilities and the ability to follow user instructions and intentions, leading to the emergence of intelligent AI assistants such as ChatGPT. However, there exists a multitude of information forms beyond language, such as vision, sound, and various sensory inputs. Visual information, in particular, is ubiquitous in the world and plays a vital role in shaping our perception and experiences. Unfortunately, current LLMs face challenges in effectively capturing, processing, and comprehending this rich visual information. To endow large models with the ability to comprehend visual features, the research community has proposed several multimodal large models Alayrac et al. (2022); Li et al. (2023a); Zhu et al. (2023), exploring the integration of a pre-trained visual encoder with a LLM to enable the unified processing of visual and textual information. For instance, LLaVA Liu et al. (2023c) introduces a shallow linear layer to align visual and textual modalities, leveraging carefully designed image-text data to pioneer instruction-following in visual-language tasks using multimodal large language models (MLLMs). In a similar vein, BLIP-2 Li et al. (2023a) proposes a lightweight Q-Former structure as a bottleneck to connect pre-trained visual encoders and LLMs. It incorporates cross-modal objectives to enhance the alignment of visual and textual features. Building upon this framework, MiniGPT-4 Zhu et al. (2023) and InstructBLIP Dai et al. (2023) adopt the architecture of BLIP-2 and achieve remarkable success in instruction-following tasks. Inspired by these achievements, some works Muhammad Maaz & Khan (2023); Li et al. (2023b); Zhang et al. (2023a); Lin et al. (2023); Li et al. (2023c) delve into extending these advancements to the domain of video. Different from images, videos often offer richer visual and temporal information, making it a challenge to explore how to better utilize the temporal cues in videos. To address this challenge, Video-ChatGPT Muhammad Maaz & Khan (2023) introduces average pooling in both temporal and spatial dimensions. As shown in fig. 1(a), it concatenates the temporal and spatial embeddings to

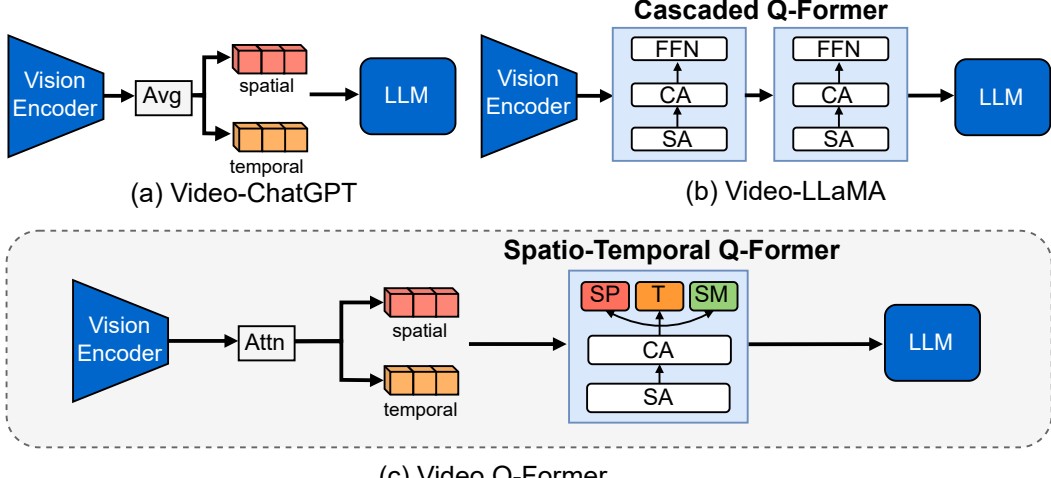

Figure 1: Comparison of Video-ChatGPT, Video-LLaMA, and Video Q-Former. (a) Video-ChatGPT utilizes a straightforward average pooling technique in both temporal and spatial dimensions. Avg means average pooling. (b) Video-LLaMA employs two cascaded Q-Former modules to extract video representations. SA refers to the self-attention layer, while CA denotes the cross-attention layer. (c) Video Q-Former introduces an attentive module to adaptively extract spatiotemporal features from videos and incorporate a spatio-temporal Q-Former to extract semantic-aligned video representations. SP, T, and SM represent the three video experts, namely SP-FFN, T-FFN, and SM-FFN, respectively.

form video representations as inputs to the LLM. This approach explicitly captures spatiotemporal features, enhancing the LLM's understanding of temporal and spatial information in videos. However, this method is limited by its inability to model temporal relationships between frames, which hinders the representation of inter-frame information. Additionally, these video representations lack semantic alignment, and may not provide adequate semantic context, making it challenging for LLMs to learn visual-language alignment. In contrast, Video-LLaMA Zhang et al. (2023a) proposes a novel approach for video representation extraction using two cascaded Q-Formers based on BLIP-2 Q-Former. Specifically, Video-LLaMA utilizes the first Q-Former to encode each frame into a set of tokens individually and subsequently employs the second Q-Former to extract video representations across all frames. This approach preserves visual feature semantics through the application of Q-Formers. However, the lack of explicit modeling of spatiotemporal features makes it challenging for LLMs to fully comprehend video features. Furthermore, employing two cascaded Q-Formers significantly diminishes the tokens available for representing video features, resulting in information loss.

In light of these concerns, we propose Video Q-Former, a novel approach that effectively tackles the aforementioned challenges by adaptively extracting spatiotemporal features from videos using an attentive module that takes into account their temporal characteristics. Additionally, we employ a spatio-temporal Querying Transformer to extract semantic-aligned video features, thereby narrowing the gap between the visual and language modalities and enhancing the LLM's comprehension of video content. By incorporating these advancements, Video Q-Former aims to construct a powerful multimodal large language modal for video understanding.

Overall, the main contributions are summarized as follows:

- We propose Video Q-Former, a multimodal large language model that exhibits strong performance across various video-grounded tasks, including multimodal instruction-following dialogues and zero-shot video question answering.

- We enhance the capability of multimodal LLMs to learn semantic-aligned spatiotemporal features of videos by incorporating an attentive module and spatio-temporal Q-Former. The

attentive module considers both spatial information and temporal cues, while the spatio-temporal Q-Former employs three video experts for semantic-aligned representations.

- We conducted extensive experiments on various tasks to validate the superior performance of our approach compared to existing methods.

## 2 RELATED WORK

### 2.1 LARGE LANGUAGE MODELS

Large Language Models (LLMs) based on Transformer Vaswani et al. (2017) architecture have been a topic of great interest in recent years. Several notable models like GPT-3 Brown et al. (2020), LLaMA Zhang et al. (2023b), BLOOM Workshop et al. (2022) and GLM Du et al. (2022) are introduced with great next token completion ability. And building upon these foundation LLMs, several intelligent assistant LLMs such as Vicuna Chiang et al. (2023), LLaMA-2 Touvron et al. (2023b) and ChatGLM Du et al. (2022) are proposed for open-domain dialogue scenario which are capable of generating helpful and human instruction following responses in dialogue systems.

### 2.2 VISION-LANGUAGE PRE-TRAINING

Vision-Language Pretraining (VLP) plays a crucial role in multi-modal tasks, harnessing the power of large-scale or even web-scale data to establish a foundational visual language model. Early on, CLIP Radford et al. (2021) emerged as a significant advancement, showcasing the potential of VLP in multi-modal tasks. The success of CLIP Radford et al. (2021) paved the way for further explorations in utilizing VLP-like methods for generative tasks, particularly in the context of multimodal large language models (MLLMs). Methods such as BLIP-2 Li et al. (2023a) and LLaVA-1.5 Liu et al. (2023b) also employ large-scale data for vision-language pre-training. These approaches enhance the architecture's ability to bridge the gap between vision and language, enabling more effective alignment and understanding of visual and textual modalities.

### 2.3 MULTIMODAL LARGE LANGUAGE MODELS

The advancements in large language models (LLMs) have sparked researchers' interest in leveraging LLMs as processing centers, complemented by visual models, for various visual-language tasks. GPT4Tools Yang et al. (2023), for instance, combines GPT-4 OpenAI (2023) with multiple vision expert models to perform visual language tasks without directly inputting features into the LLM. Other approaches, such as LLaVA Liu et al. (2023c), employ a simple linear layer to connect a visual encoder with LLMs. Similarly, works like BLIP-2 Li et al. (2023a), InstructBLIP Dai et al. (2023), and MiniGPT-4 Zhu et al. (2023) utilize a lightweight Q-Former architecture to extract valuable information from original features generated by a Vision Transformer Dosovitskiy et al. (2020), enhancing the alignment between visual and textual features. In the domain of videos, focusing on temporal-aware architectures, Video-ChatGPT Muhammad Maaz & Khan (2023) adopts average pooling to capture both temporal and spatial information in videos, coupled with video instruction-following data, leading to significant achievements in the field of video understanding. Video-LLaMA Zhang et al. (2023a) introduces the cascaded Q-Former structure to extract spatiotemporal information from videos. However, Video-ChatGPT has limitations in modeling temporal relationships between frames, while Video-LLaMA does not explicitly model spatiotemporal features, both of which are essential for a comprehensive understanding of videos.

## 3 METHOD

### 3.1 MODEL ARCHITECTURE

Given Q-Former's promising capability to extract semantic-aligned vision features, it is feasible to encode each frame into 32 queries individually, and directly feed to LLM with the concatenated query embeddings for video-text tasks. However, this method results in significant computational overhead due to the increased input sequence length of the LLM. As demonstrated in table 1, when sampling 16 frames from videos, Q-Former exhibits twice the FLOPs compared to our proposed

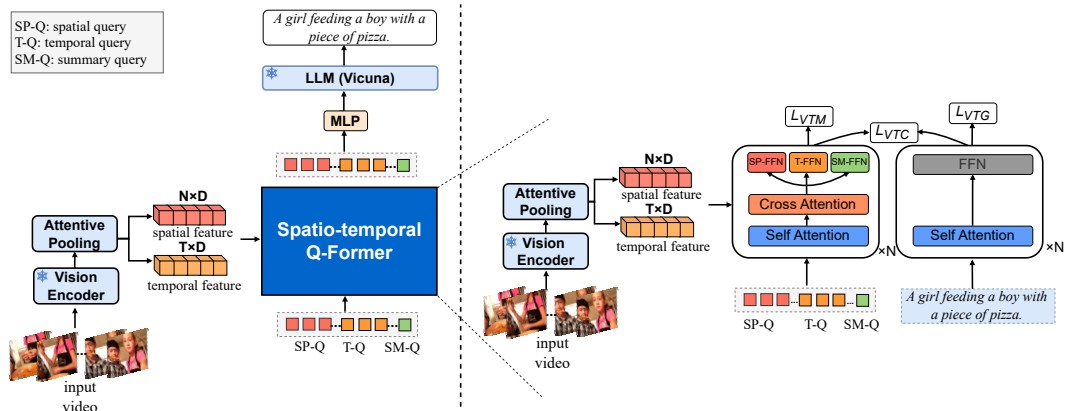

Figure 2: The pipeline of our method and the details of spatio-temporal Q-Former. **Left:** Overall pipeline of our method. Input videos are encoded into spatial features and temporal features by the image encoder and attentive pooling module. Subsequently, the spatio-temporal Q-Former extracts language-informative video embeddings, and LLM (Vicuna) decodes texts conditioned on the input video embeddings. **Right:** Details of spatio-temporal Q-Former and first-stage pre-training objectives. Spatio-temporal Q-Former comprises a text transformer and an MoE image transformer with three video experts: SP-FFN, T-FFN, and SM-FFN. In the first stage of pre-training, we jointly optimize the video-text matching loss (VTM), video-text contrastive learning loss (VTC), and video-grounded text generation loss (VTG) to enable the spatio-temporal Q-Former in extracting language-informative video representations.

spatio-temporal Q-Former. Furthermore, this method does not incorporate spatiotemporal modeling of videos, which poses challenges for large language models in comprehending videos. To mitigate the aforementioned issues, we propose Video Q-Former, a novel multimodal large language model for video understanding. Video Q-Former employs a spatio-temporal querying transformer to extract language-informative spatial and temporal features from videos concurrently. This querying transformer serves as a bottleneck module that connects the frozen vision encoder and LLM. Our model, as depicted in fig. 2 (right), consists of an attentive pooling module and a spatio-temporal Q-Former, which includes a MoE image transformer and a text transformer.

**Attentive Pooling Module** The explicit modeling of spatiotemporal features in videos is crucial for enabling large language models to understand video content effectively. It allows LLMs to capture rich semantic information, dynamic changes, and contextual cues, thereby enhancing their ability to comprehend the content of videos. Hence, we devise an attentive pooling module to learn decoupled spatiotemporal features of videos. The attentive pooling module consists of a cross attention layer and feed-forward layer that allow for the acquisition of spatiotemporal representations of videos through a learnable pooling process. Given an input video $\mathbf{V} \in \mathbb{R}^{T \times H \times W \times C}$ with $T$ frames, where $H$, $W$, $C$ represent the frame height, frame width, and frame channel, respectively. We employ a vision encoder to individually encode the frames into vision representations, resulting in an initial video embedding $x \in \mathbb{R}^{T \times N \times D}$. Here, $N$ denotes the number of patches per frame, and $D$ represents the feature dimension. To extract spatio-temporal features from the video, we introduce two attentive pooling queries that learn to extract the corresponding features. Specifically, we utilize a learnable spatial pooling query $Q_s \in \mathbb{R}^{1 \times D}$ to cross-attend to video embedding $x \in \mathbb{R}^{T \times N \times D}$ and generate temporal representation $v_t \in \mathbb{R}^{T \times D}$. Similarly, a learnable temporal pooling query $Q_t \in \mathbb{R}^{1 \times D}$ is employed to cross-attend to transposed video embedding $x' \in \mathbb{R}^{N \times T \times D}$ and yield spatial representation $v_s \in \mathbb{R}^{N \times D}$. This process is formulated as follows:

$$\tilde{v}_t = CA(Q_s, x, x), \tag{1}$$
$$v_t = FFN(\tilde{v}_t) + \tilde{v}_t, \tag{2}$$
$$\tilde{v}_s = CA(Q_t, x', x'), \tag{3}$$
$$v_s = FFN(\tilde{v}_s) + \tilde{v}_s, \tag{4}$$

Table 1: Comparsion of FLOPs. "ST Q-Former" means spatio-temporal Q-Former. 16 frames are sampled to calculate FLOPs

| Method | Q-Former | ST Q-Former | BLIP-2 | Video Q-Former |
|--------|----------|-------------|--------|----------------|
| FLOPs  | 102.1G   | **52.6G**   | 11.3T  | **8.57T**      |

**SP-Feature**: spatial feature      **SP-Q**: spatial query positions

**T-Feature**: temporal feature      **T-Q**: temporal query positions

☐ unmasked     ☐ masked      **SM-Q**: summary query positions

Figure 3: The cross-attention mask we used to control the visibility of spatial and temporal queries on various spatiotemporal features. Spatial queries and temporal queries are limited to attending only to their corresponding features, while the summary query can attend to all features

where $CA(q, k, v)$ represents the cross-attention layer, and $FFN(\cdot)$ denotes the feed-forward network. By implementing a learnable attentive pooling process, we can acquire decoupled video features, thereby taking into account both the temporal relationship between frames and the spatial characteristics within each frame.

**Spatio-temporal Q-Former** The spatio-temporal Q-Former is designed to connect the vision encoder and LLM, bridging the gap between vision and language modalities. It consists of a MoE image transformer and a text transformer. The MoE image transformer, drawing inspiration from previous works Bao et al. (2022); Shazeer et al. (2017); Gan et al. (2020), replace the feed forward network with a mixture of video experts. Specifically, three video experts are introduced: the spatial video expert (SP-FFN), the temporal video expert (T-FFN), and the summary video expert (SM-FFN). To extract language-informative spatiotemporal visual representations, a set number of learnable spatial query, temporal query, and summary query are employed. In our experiment, we employed one query for summarization and 64 queries for the extraction of spatial and temporal features. Similar to BLIP-2, these queries interact with each other through self-attention layers and with the decoupled video representations through cross-attention layers. Moreover, the queries can also interact with the text through the same self-attention layers. All different queries share the same self-attention layer and cross-attention layer. To ensure that queries cross-attend to the corresponding video representations, a cross-attention mask is designed, as depicted in fig. 3. Ultimately, different experts are employed to process different query embeddings in parallel.

### 3.2 MODEL PRE-TRAINING

To accomplish the goal of extracting semantic-aligned spatiotemporal features from videos, we employ a two-stage pre-training approach and apply the identical self-attention mask strategy as BLIP-2. In the first stage, the spatio-temporal Q-Former is trained to extract spatio-temporal video embeddings that are most relevant to the text. This training process consists of jointly optimizing three losses: Video-Text Contrastive Learning (VTC) loss, Video-grounded Text Generation (VTG) loss, and Video-Text Matching (VTM) loss. The VTC and VTG losses resemble the corresponding losses in BLIP-2. Regarding the VTM loss, we compute the average of the spatial queries and temporal queries separately. Subsequently, we concatenate the averaged queries with the [CLS] to-

ken and input them into a binary classification task to predict whether the video-text pair is match or not. In the second pre-training stage, we connect the spatio-temporal Q-Former with the vision encoder to a frozen LLM. To accomplish this, we use a two-layer MLP to project the output query embeddings from the spatio-temporal Q-Former to the LLM embedding space. The spatio-temporal Q-Former is then adapted to the frozen LLM through language modeling loss conditioned on the query embeddings.

## 4 EXPERIMENT

### 4.1 PRE-TRAINING SETUP

Table 2: **Performance comparison on zero-shot video question answering on Video-ChatGPT benchmark Muhammad Maaz & Khan (2023).** Video Q-Former achieves state-of-the-art performance across various datasets.

| Method | MSVD-QA | | MSRVTT-QA | | TGIF-QA | | ActivityNet-QA | |
|---|---|---|---|---|---|---|---|---|
| | Accuracy | Score | Accuracy | Score | Accuracy | Score | Accuracy | Score |
| FrozenBiLM | 32.2 | - | 16.8 | - | 41.0 | - | 24.7 | - |
| Video-LLaMA | 51.6 | 2.5 | 29.6 | 1.8 | - | - | 12.4 | 1.1 |
| LLaMA-Adapter | 54.9 | 3.1 | 43.8 | 2.7 | - | - | 34.2 | 2.7 |
| VideoChat | 56.3 | 2.8 | 45.0 | 2.5 | 34.4 | 2.3 | 26.5 | 2.2 |
| Video-ChatGPT | 64.9 | 3.3 | 49.3 | 2.8 | 51.4 | 3.0 | 35.2 | 2.7 |
| BT-Adapter | 67.5 | 3.7 | 57.0 | 3.2 | - | - | 45.7 | 3.2 |
| Valley-v3 | 60.5 | 3.3 | 51.1 | 2.9 | - | - | 45.1 | 3.2 |
| Video Q-Former | **77.4** | **3.8** | **70.1** | **3.4** | **69.0** | **3.7** | **47.1** | **3.2** |

**Pre-training Datasets**   We employ a combination of three public datasets to pre-train our model. These datasets are WebVid-2M Bain et al. (2021), Conceptual Captions 3M Sharma et al. (2018), and InternVid Wang et al. (2023), comprising a total of approximately 15 million examples. (a) WebVid-2M Bain et al. (2021) is a large-scale dataset that contains video-text pairs scraped from stock footage websites. (b) Conceptual Captions 3M Sharma et al. (2018) is an extensive image caption dataset that we use as static video during the pre-training phase of our model. (c) InternVid Wang et al. (2023) is a newly proposed dataset that consists of 10 million video clips accompanied by high-quality generated captions.

**Implementation Details**   We implement our method based on BLIP-2 and initialize the model from it. The frozen image encoder we utilized is ViT-G/14 from EVA-CLIP Fang et al. (2023). For the frozen language model, we employ Vicuna Chiang et al. (2023), an open-source chatbot fine-tuned on LLaMA Touvron et al. (2023a). To process videos, we employ a frame sampling rate of 1 FPS and utilize 8 frames for pre-training and 16 frames for downstream tasks. In the first stage, we pre-train for 58k steps with a batch size of 100, using a dataset ratio of 1:1:2. Notably, to better leverage the pre-trained parameters of BLIP-2, we choose to freeze the self-attention and cross-attention layers during this stage, while unfreezing the video experts, namely SP-FFN, T-FFN, and SM-FFN. In the second stage, we pre-train for 220k steps with a batch size of 16, maintaining the same dataset ratio. At this stage, we unfreeze all parameters of the spatio-temporal Q-Former. The first stage took 4 days, while the second stage took 5 days when performed on two 8-A100(80G) machines. To improve the ability to follow instructions, we conduct 15 epochs of training using 100,000 high-quality video instruction data collected by Video-ChatGPT Muhammad Maaz & Khan (2023). For optimization, we used AdamW Loshchilov & Hutter (2017) with $\beta_1 = 0.9$ and $\beta_2 = 0.999$, along with a weight decay of 0.01. Our learning rate schedule incorporated a linear decay, starting with a peak learning rate of 1e-4 in the first stage and 4e-5 in the second stage. We applied a linear warmup for the first 2k steps and then decayed the learning rate to zero. The image scale used is 224×224, and horizontal flipping is employed as data augmentation.

### 4.2 MULTI-MODAL INSTRUCTION FOLLOWING

Vicuna Chiang et al. (2023) exhibits remarkable proficiency in instruction-following dialogues, but it lacks the capacity to process and comprehend multimodal content. By effectively extracting video

Table 3: **Performance of video-based text generation.** "V-" in the model names stands for "Video-" and "LLaMA-Ada" stands for method LLaMA-Adapter. The higher the values of these metrics, the better the performance.

| Evaluation Aspect | VideoChat | V-ChatGPT | LLaMA-Ada | V-LLaMA | Valley-v3 | Video Q-Former |
|---|---|---|---|---|---|---|
| Correctness of Information | 2.25 | 2.50 | 2.03 | 1.96 | 2.43 | **2.74** |
| Detail Orientation | 2.50 | 2.57 | 2.32 | 2.18 | 2.13 | **2.67** |
| Contextual Understanding | 2.54 | 2.69 | 2.30 | 2.16 | 2.86 | **2.97** |
| Temporal Understanding | 1.98 | 2.16 | 1.98 | 1.82 | 2.04 | **2.49** |
| Consistency | 1.84 | 2.20 | 2.15 | 1.79 | 2.45 | **2.82** |

features and enabling Vicuna to understand visual content, Video Q-Former empowers it to engage in video-based multimodal dialogues. In Fig. 4, we showcase several examples that demonstrate the impressive video-based multimodal instructed dialogue capability of our model. Additional qualitative results can be found in the appendix.

### 4.3 ZERO-SHOT VIDEO QUESTION ANSWERING

We utilize the benchmarks established by Video-ChatGPT Muhammad Maaz & Khan (2023) to evaluate zero-shot video question answering capability. It employs ChatGPT to assess the accuracy of the model's prediction results and assigns a score ranging from 0 to 5 points to indicate the degree of meaningful correspondence. Please refer to the supplementation materials for the evaluation prompt. We conduct experiments on MSVD-QA Chen & Dolan (2011), MSRVTT-QA Xu et al. (2017), TGIF-QA Li et al. (2016) and ActivityNet-QA Yu et al. (2019). The results are shown in table 2. Video Q-Former consistently outperforms other methods by a large margin and achieves state-of-the-art across all video question answering datasets. Our approach yields the best outcomes, achieving 70.1% accuracy on MSRVTT-QA and 77.4% accuracy on MSVD-QA. This represents an improvement over the second-best model by nearly 13% and 10%, respectively. Furthermore, when compared to VideoChat Li et al. (2023b), which employs a video encoder and Q-Former structure and conducts pre-training on a larger dataset, Video Q-Former achieves a nearly 1-point score increase in both MSVD-QA and MSRVTT-QA. These results demonstrate Video Q-Former's superiority of video understanding and accurate language generation ability.

Table 4: **Performance comparison on video captioning.** B@4: BLEU@4. Video Q-Former achieves competitive results compared to other methods, despite being pre-trained on a smaller amount of data.

| Method | #PT Data | MSRVTT | | | MSVD | | |
|---|---|---|---|---|---|---|---|
| | | B@4 | CIDEr | Rouge-L | B@4 | CIDEr | Rouge-L |
| UniVL | 136M | 42.2 | 49.9 | 61.2 | - | - | - |
| SwinBERT | - | 41.9 | 53.8 | 62.1 | 58.2 | 120.6 | 77.5 |
| CLIP4Caption | - | 46.1 | 57.7 | 63.7 | - | - | - |
| MV-GPT | 69M | 48.9 | 60.0 | 64.0 | - | - | - |
| HiTeA | 17M | 49.2 | 65.1 | 65.0 | 71.0 | 146.9 | 81.4 |
| VideoCoCa | 3B | 53.8 | 73.2 | 68.0 | - | - | - |
| GIT | 0.8B | 53.8 | 73.9 | 67.7 | 79.5 | 180.2 | 87.3 |
| GIT2 | 12.9B | **54.8** | **75.9** | **68.2** | **82.2** | **185.4** | **88.7** |
| Video Q-Former | 15M | 51.2 | 70.8 | 66.5 | 76.3 | 174.2 | 86.3 |

### 4.4 VIDEO-BASED TEXT GENERATION PERFORMANCE BENCHMARKING

To assess the video-based text generation performance of Video Q-Former, we will evaluate five key aspects: Correctness of Information, Detail Orientation, Contextual Understanding, Temporal Understanding, and Consistency. The table 3 presents the results of the evaluation. Based on the findings, Video Q-Former outperforms other methods in all five aspects, establishing new state-of-the-art records in this benchmark, and demonstrating an improved comprehension of videos along with the capacity to generate high-quality text. Particularly, in terms of Temporal Understanding, our method surpasses the second-ranking method, Video-ChatGPT Muhammad Maaz & Khan (2023),

which incorporates average pooling along the spatial and temporal dimensions, by 0.33 points. This underscores the efficacy in capturing the temporal characteristics of videos through our proposed attentive module and spatio-temporal Q-Former.

## 4.5 VIDEO CAPTIONING

We compared with other vision-language pre-training models on several video captioning datasets, including MSRVTT Xu et al. (2016), MSVD Chen & Dolan (2011). The results of these comparisons are presented in table 4. Note that we do not use any CIDEr optimization method such as SCST Rennie et al. (2017). Remarkably, even with less pre-training data, Video Q-Former achieved competitive performance when compared to VideoCoCa Yan et al. (2022) and GIT Wang et al. (2022), exemplifying the training efficiency of our method. Moreover, our method significantly outperformed HiTeA Ye et al. (2022), a model that utilizes a similar amount of pre-training data, demonstrating the strong video-to-text generation capabilities of Video Q-Former.

Table 5: **Results for video summarization.** R: ROUGE-L, C: CIDEr. Video Q-Former significantly outperforms other methods on the Video-CSR dataset, establishing new state-of-the-art records.

| Model | #PT Data | BLEURT | R | C |
|---|---|---|---|---|
| VideoCoCa | 0.5M | 36.8 | 22.4 | 9.5 |
| Video-Teller | 0.5M | 47.1 | 23.5 | 11.2 |
| Video Q-Former | - | **56.7** | **32.1** | **33.4** |

## 4.6 VIDEO SUMMARIZATION

To comprehensively evaluate the video-to-text generation ability of long-form videos, we conduct experiments on video summarization task. The objective of the video summarization task is to generate summaries that effectively capture the essence of the video while including more detailed information. This task presents a challenge as it requires the model to generate longer and more descriptive captions. To benchmark our results, we utilize the Video-CSR dataset proposed by Liu et al. (2023d), which is designed for long-form video understanding. This dataset consists of 4.8 thousand video clips carefully chosen from previously published YouTube-based video datasets Abu-El-Haija et al. (2016); Zellers et al. (2022). The video clips varies in content and length, ranging from a few seconds to one minute. Each video clip is accompanied by 5 concise captions and 5 extensive summaries, all of which are human-annotated. Moreover, the dataset provides rich ASR texts. ASR is crucial in capturing detailed information from the video, and hence, we utilizes it as a text prompt in our experiment. Specifically, we append the ASR texts to the query embedding and input the combined prompt to the LLM. The LLM is then prompted to generate detailed video summary. As shown in table 5, experimental results demonstrates that Video Q-Former achieves remarkable performance in video summarization. Our method significantly surpasses the VideoTellerLiu et al. (2023a) method by approximately 10 points in terms of BLEURT Sellam et al. (2020), a semantic-based evaluation metric that is well-suited for the evaluation of lengthy texts Liu et al. (2023a). The generated summaries successfully capture the main events and details of the videos while maintaining coherence and relevance. This can be attributed to the effective spatiotemporal modeling of videos by Video Q-Former. Notably, our method achieves state-of-the-art performance in the video summarization task even without pre-training on the 0.5 million video-text pairs mentioned by Video-Teller Liu et al. (2023a).

## 4.7 ABLATION STUDY

**Effects of key components** The effectiveness of critical components is demonstrated in table 6. For spatiotemporal pooling, we adopt the average pooling approach proposed by Video-ChatGPT Muhammad Maaz & Khan (2023) as the baseline. For Q-Former, we use the original Q-Former from BLIP-2Li et al. (2023a) with an equal number of queries as our baseline method. To perform a comparison between different variants of the model, we conduct pre-training on WebVid-2M Bain et al. (2021) for a total of five epochs and finetune on MSRVTT Xu et al. (2016) caption dataset for one epoch. Our experimental findings indicate that attentive pooling module outperforms average pooling, which fails to consider the temporal relations among video frames. Furthermore,

Table 6: **Ablation studies of key components on the video captioning datatset MSRVTT.** Different spatio-temporal pooling methods are indicated by Average Pooling and Attentive Pooling, respectively. SP Q-Former represents our proposed spatio-temporal Q-Former.

| Average Pooling | Attentive Pooling | Q-Former | SP Q-Former | B@4 | CIDEr | ROUGE-L |
|:---:|:---:|:---:|:---:|:---:|:---:|:---:|
| ✓ | | ✓ | | 47.76 | 62.74 | 64.61 |
| ✓ | | | ✓ | 47.59 | 62.92 | 64.78 |
| | ✓ | ✓ | | 49.62 | 66.68 | 65.86 |
| | ✓ | | ✓ | **50.15** | **67.05** | **66.03** |

Table 7: **Ablation study of spatial and temporal video experts.**

| Method | ActivityNet-QA | Temporal Understanding |
|:---|:---:|:---:|
| Spatial only | 3.22 | 2.45 |
| Temporal only | 3.36 | 2.57 |
| Video Q-Former | **3.41** | **2.58** |

Table 8: **Ablation study of spatial and temporal video experts** on the video captioning dataset MSRVTT.

| | MSRVTT | | |
|:---|:---:|:---:|:---:|
| Method | B@4 | CIDEr | Rouge-L |
| Temporal only | 47.76 | 62.20 | 64.90 |
| Spatial only | 50.08 | 66.98 | 65.88 |
| Video Q-Former | **50.15** | **67.05** | **66.03** |

the spatio-temporal Q-Former achieves superior performance compared to the original Q-Former, thus validating the superiority of our method.

**Effectiveness of video experts** We performed an ablation study of the video experts on various benchmarks, as shown in table 7 and table 8. The videos in ActivityNet-QA Yu et al. (2019) have an average duration of 3 minutes, longer than most VideoQA benchmarks such as MSRVTT-QA Chen & Dolan (2011), where videos average 15 seconds. In addition, the Temporal Understanding metric introduced in Video-ChatGPT Muhammad Maaz & Khan (2023) assesses a model's ability to understand temporal aspects. Therefore, a more profound comprehension of temporal characteristics is crucial for these evaluation metrics. The results presented in table 7 highlight the significance of grasping temporal information for precise comprehension. On the other hand, the MSRVTT video captioning task focuses more on spatial details. The findings in table 8 demonstrate that spatial features play a crucial role in generating accurate captions. Furthermore, the collaboration of spatial and temporal video experts in Video Q-Former achieves the highest performance both in table 7 and table 8, thus emphasizing the superiority of explicit spatiotemporal video representations in video understanding.

## 5 CONCLUSION

In this paper, we propose Video Q-Former, a multimodal large language model for video understanding that utilizes an attentive module and a spatio-temporal querying transformer. Our attentive module is designed to adaptively extract spatiotemporal features from videos, considering both spatial information within frames and temporal dynamics between frames. The spatio-temporal Q-Former incorporates three video experts, namely SP-FFN, T-FFN, and SM-FFN, to simultaneously extract semantic-aligned spatial and temporal video representations. This enhancement greatly improves the comprehension of video content by the LLM. Extensive experiments demonstrate that our model achieves state-of-the-art performance on zero-shot video question answering datasets and delivers competitive results on other tasks.

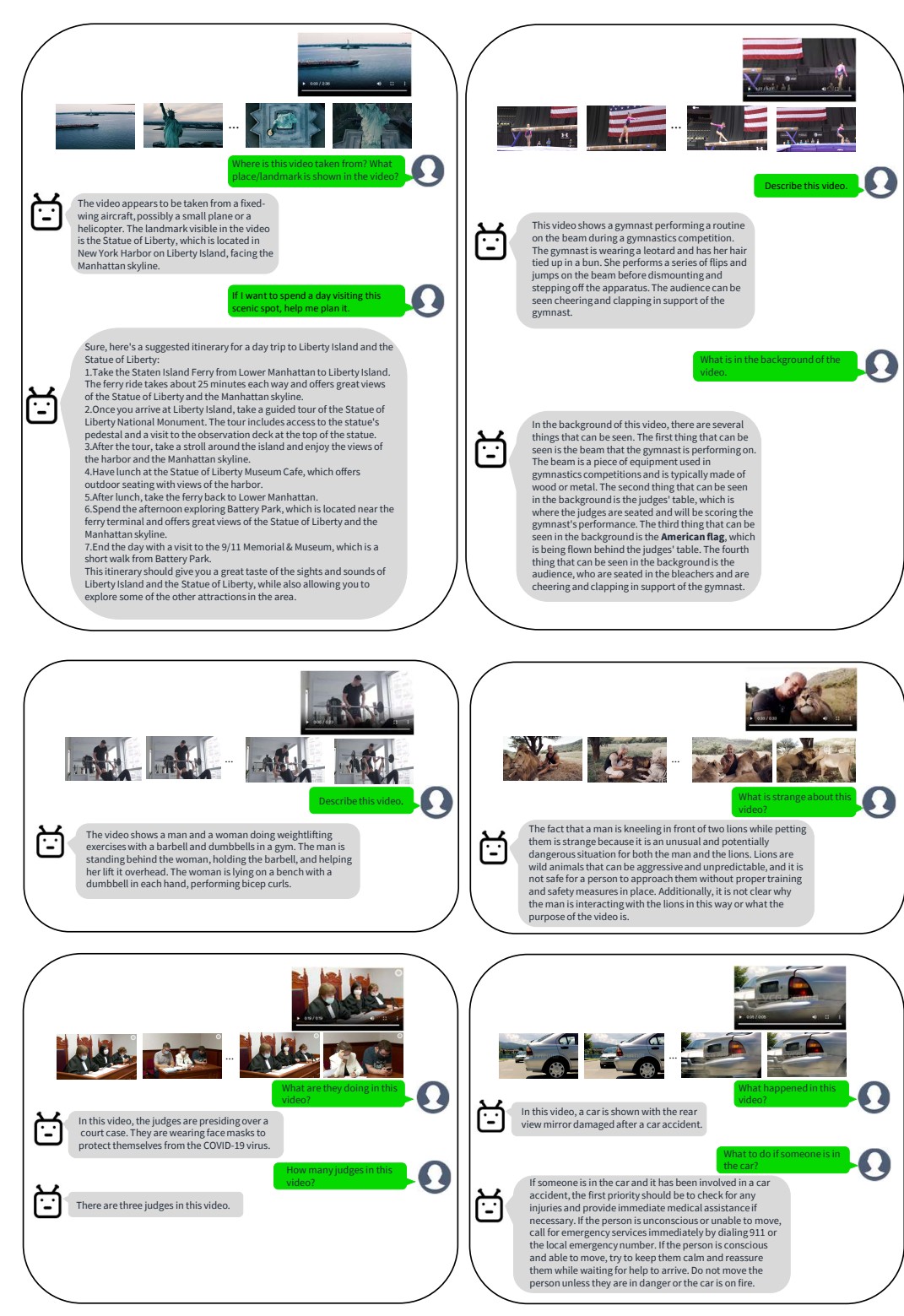

Figure 4: Examples of multimodal instructed zero-shot video-to-text generation demonstrate the capabilities of Video Q-Former, including video-based visual conversation and visual knowledge reasoning.

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

# A  APPENDIX

# B  MORE ABLATION STUDY

In this section, we further perform ablation studies on our method. All experiments are pre-trained on WebVid-2M Bain et al. (2021) and subsequently finetuned on the MSRVTT video captioning dataset Xu et al. (2016), unless otherwise mentioned.

Table 9: **Ablation study of the number of queries** on MSRVTT video captioning.

| #queries | MSRVTT | | |
|---|---|---|---|
| | B@4 | CIDEr | Rouge-L |
| 32 | 47.48 | 62.50 | 64.72 |
| 64 | **50.15** | 67.05 | **66.03** |
| 128 | 49.80 | **67.16** | 65.88 |

**Analysis on the number of queries**   In this section, we evaluate the influence of the number of query tokens on the results. In our method, we set the query size for spatial and temporal queries to be the same. Specifically, we compare the performance of three different query sizes: 32, 64, and 128. As shown in Tab. 9, using only 32 queries is insufficient to capture all the necessary video information. Conversely, using 128 queries yields the best results on the CIDEr Vedantam et al. (2015) metric but also incurs excessive computational costs. By using 64 queries, we achieve a trade-off between capturing sufficient information and minimizing computational costs.

Table 10: **Ablation study of spatio-temporal Q-Former** on MSRVTT video captioning. "Attn only" represents the use of attentive pooling only, without the spatio-temporal Q-Former. Note that Video Q-Former consists of both attentive pooling and spatio-temporal Q-Former.

| Method | MSRVTT | | |
|---|---|---|---|
| | B@4 | CIDEr | Rouge-L |
| Attn only | 40.62 | 56.68 | 60.54 |
| Video Q-Former | **50.15** | **67.05** | **66.03** |

**Analysis of spatio-temporal Q-Former**   In this study, we conduct an ablation study to examine the effects of the spatio-temporal Q-Former. We compared the performance of the model with and without the presence of the spatio-temporal Q-Former. The results, presented in Tab. 10, clearly indicate that using only attentive pooling alone does not yield satisfactory results. However, when combined with the spatio-temporal Q-Former, our method achieves better results. These findings not only validate the effectiveness of the spatio-temporal Q-Former, but also demonstrate that the Q-Former structure facilitates the extraction of semantic-aligned video representations and enhances the LLM's understanding of videos in a more efficient manner.

**Effects of MLP layer**   In the context of self-supervised learning, the use of an MLP projection has been found to outperform a linear layer Chen et al. (2020). In order to investigate the impact of the choice of projector, we conducted ablation experiments. The results are displayed in table 11. Significantly, when the MLP layer is omitted, the performance on the CIDEr metric experiences a notable decline of 3 points, as evidenced by the experiments conducted on the MSVD dataset. These findings underscore the superiority of utilizing a two-layer MLP within the context of Video Q-Former  for video captioning tasks. Moreover, the incorporation of a two-layer MLP effectively facilitates Video Q-Former  in acquiring more robust multimodal representations, thus validating its efficacy in improving the learning process.

Table 11: **Ablation study of MLP** on MSRVTT and MSVD video captioning.

| Method | MSRVTT | | | MSVD | | |
|---|---|---|---|---|---|---|
| | B@4 | CIDEr | Rouge-L | B@4 | CIDEr | Rouge-L |
| Video Q-Former w/o MLP | 50.27 | 68.45 | 66.15 | 74.64 | 165.92 | 85.04 |
| Video Q-Former | **50.72** | **69.33** | **66.40** | **75.72** | **168.79** | **86.12** |

## C  QUALITATIVE RESULTS

The qualitative results, illustrating the range of capabilities of the Video Q-Former, are presented in Fig. 5. These examples depict various tasks performed by Video Q-Former such as video conversation, video reasoning, creative and generative tasks, action recognition, etc.

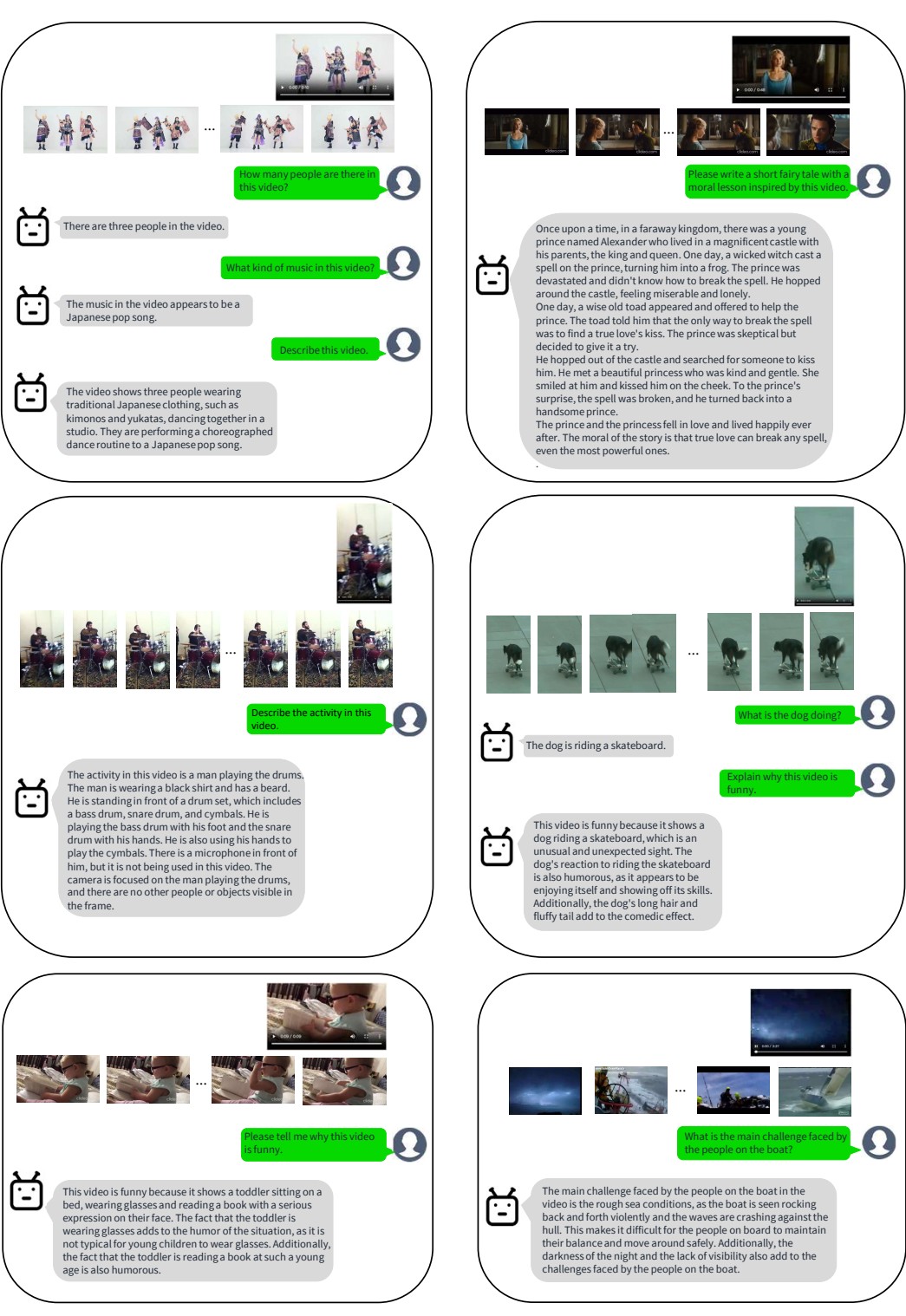

Figure 5: Selected examples of various tasks performed by Video Q-Former.

