# OpenReview forum: "Video Q-Former: Multimodal Large Language Model with Spatio-Temporal Querying Transformer Towards Video Understanding"
_ICLR.cc/2025/Conference — Submitted to ICLR 2025_

### Official Review · Reviewer_uaDg · 2024-10-20

**Soundness:** 3
**Presentation:** 2
**Contribution:** 2
**Rating:** 5
**Confidence:** 4

**Summary:**

This paper proposes Video-Qformer, a connection module that extracts spatiotemporal features from videos to enhance large language models (LLMs). The Video-Qformer architecture consists of an Attentive Pooling Module and a Q-Former with expert feed-forward networks (FFNs) specialized for spatial, temporal, and summarization queries. The model is evaluated on several benchmarks, including zero-shot VideoQA, Video-ChatGPT, video captioning, and video summarization, with ablation studies demonstrating the effectiveness of individual components.

**Strengths:**

1) This paper introduces an Attentive Pooling Module that can decouple the spatiotemporal features of videos with cross-attention layers, which differs from the average pooling in Video-ChatGPT.
2) Three FNN experts are introduced into the Q-former to handle the spatial, temporal, and summarization queries respectively.
3) Ablation studies are conducted to verify each component of the Video-Qformer.

**Weaknesses:**

1) The introduction section (lines 30–50) spends excessive space on general background information that is only tangentially relevant to the main contribution—the design of an efficient video connection module for MLLMs. This could be streamlined for clarity.
2) The paper does not adequately explain the motivation behind the Mixture-of-Experts (MoE) design or the rationale for decoupling spatiotemporal features through attentive pooling. Adding visualizations (e.g., attention maps) and deeper analysis of these choices would strengthen the work.
3) This paper doesn't provide further insights or solve the main problems of current video MLLMs since it uses a setting of 8/16 frames for video understanding. If each frame is encoded as 32 tokens (just as what BLIP-2 did), 16 frames only constitute 512 tokens. Video representations with such a token length can be well accepted by almost all LLMs nowadays, there is no need to compress them further. Sometimes even an image-based MLLM will use 576 tokens to represent one single image (e.g., LLaVA-1.5).
4) The experimental design raises concerns about fairness. The main comparison targets (Video-LLaMA and Video-ChatGPT) use much less video data to train their connectors, which complicates direct performance comparisons and raises doubts about the improvements of Video-Qformer.

**Questions:**

See Weaknesses

---

### Official Review · Reviewer_NyYZ · 2024-10-30

**Soundness:** 2
**Presentation:** 1
**Contribution:** 2
**Rating:** 3
**Confidence:** 4

**Summary:**

In this paper, the authors propose the Video Q-Former model, a large multimodal language model for video understanding that leverages an attentive module and spatio-temporal query transformers. Compared to previous video understanding methods, it adaptively extracts spatio-temporal features from videos while considering spatial information within frames and temporal dynamics between frames.

**Strengths:**

Video Q-Former adaptively extracts spatiotemporal features from videos while considering spatial information within frames and temporal dynamics between frames. The Spatiotemporal Q-Former integrates three video experts, namely SP-FFN, T-FFN, and SM-FFN, to simultaneously extract semantically aligned spatial and temporal video representations, achieving better performance.

**Weaknesses:**

- The authors claim that their method achieves state-of-the-art performance. However, the comparison methods lack recent works, such as VideoChat2, LLaVA-OneVision, MiniCPMv, and PLLaVA. The related work section also lacks recent works.

- There is a lack of novelty. The proposed Attentive Pooling Module has been widely used in previous works, such as [1].

[1] Temporal-attentive Covariance Pooling Networks for Video Recognition. NeurIPS 2021.

- It is recommended that the authors visualize the results of the Attentive Pooling Module to better demonstrate the effectiveness of this module.

- Video-ChatGPT and Video-LLaMA are the main comparison methods, as the Video Q-Former can be considered a combination of the two. However, in the experiments, the authors did not clarify whether all models were trained on the same dataset when comparing Video-LLaMA and Video-ChatGPT. This raises confusion as to whether the performance advantage of Video Q-Former comes from the model architecture or the dataset.

- The results in Table 6 show that the improvement brought by SP Q-Former is quite limited. For example, in terms of the CIDEr metric, it only improved by 0.37. The ablation experiment in Table 8 also shows that compared to "Spatial only," the improvement of "Video Q-Former" is also quite limited, improving by only 0.07 on the CIDEr metric.

**Questions:**

See weakness.

---

### Official Review · Reviewer_Ccfd · 2024-11-04

**Soundness:** 2
**Presentation:** 3
**Contribution:** 2
**Rating:** 3
**Confidence:** 4

**Summary:**

This paper proposes the Video Q-Former for video-based large language models (LLMs). The proposed module is designed to extract semantically aligned spatiotemporal features from videos, addressing limitations in existing models in capturing inter-frame relationships. Experiments across various tasks demonstrate the effectiveness of the proposed method.

**Strengths:**

- The proposed method appears to be more efficient than BLIP-2.
- The method is straightforward and easy to understand.

**Weaknesses:**

- The improvement in computational efficiency appears to stem from applying attentive pooling to the original video, whereas Video-LLaMA computes over all video tokens. This may lead to significant information loss, as many details are discarded during pooling.
- The method seems to build on Video-ChatGPT by adding a Q-Former module, which introduces three different types of query tokens. However, it lacks innovation and technical depth overall.
- In comparing with the baseline in Table 2, it is not clear if the method was trained on the same data, especially in comparison with Video-ChatGPT and Video-LLaMA. Additionally, the versions of the LLMs used should be consistent and clearly specified.
- Since Video-ChatGPT, numerous updated methods have been introduced, and this paper lacks comparisons with these baselines, such as VideoChat2, LLava-Video, Chat-Unity, and others.

**Questions:**

Please refer to the weaknesses.

---

### Official Review · Reviewer_F39o · 2024-11-04

**Soundness:** 2
**Presentation:** 3
**Contribution:** 2
**Rating:** 5
**Confidence:** 4

**Summary:**

The work is proposed to pioneer video-language understanding with large language models. Motivated by that the current large video language models have limitations in capturing inter-frame relationships, the paper proposes Video Q-Former with flexible queries. Specifically, the authors set spatial queries, temporal queries, summary queries to attend to different set of vision tokens. The model architecture design in paper is short, and most part falls in experiments. The experiments show that the method can achieve good zero-shot QA performance on main steam video question answering benchmarks. They also show promising performance on video captioning tasks, such as MSRVTT, MSVD.

**Strengths:**

1. The idea of using different temporal, spatial and summary queries for spatio-temporal representation learning is easy and straightforward. It makes sense. Also, according to Table 1, the computation cost is acceptable and not higher than BLIP-2, while achieving superior performance compared to BLIP-2.
2. The paper experiments on video captioning benchmark, zero-shot question answering benchmark and video summarization benchmarks. Some important works are included but the most recent ones are missing.
3. The writing of the paper is clear and the illustration is good.

**Weaknesses:**

1. The approach of decomposing video representation into spatial and temporal representation for efficient and effective spatio-temporal modelling is a general idea in video understanding. I'm not going to blame using this in large video language models, however, I think proper credit and literature reviews should be included. For example, TimeSFormer[1], Uniformer[2], Dual-AI[3] and others using transformer for video recognition.
2. Training specialized and efficient video QFormer has been explored and utilized by UMT [4] and VideoChat2 [5]. Please clarify the difference and include them in literature reviews.
2. Comfusing attentive pooling module architecture. It seems the temporal representation $v_{t}$ is derived from spatial queries $Q_{s}$ attending to a set of frame features (with T as batch dimension). It means the spatial queries can only attend to in-frame content. This is doubtable why the representation is called temporal representation.
3. Training data: what specific data are used from training? Please provide details of how many videos from what dataset and how you make sampling. This is critical for reproduction and measure the method effectiveness.
4. Experiments - Comparison fairness: More latest methods should be included in comparison, especially those with similar motivations, e.g., VideoChat2.
5. Experiments - Image benchmark: As image dataset is used, it would be great to show the performance variance after such ST QFormer tuning. Also compared to normal QFormer.
6. Experiments - Video summarization: there are some new good benchmarks, like Shot2Story ranging from different topics and using only text and frames modalities. This is not mandatory, but it should be good to include.
7. Experiments - Ablation - missing components: There should be experiments and explanation regarding the different queries used in spatio-temporal representation, i.e., spatial, temporal and summary. That is the key difference to VideoChatGPT and other works. What if only have spatial one, or temporal and summary one?
8. Experiments - Ablation - metric: for abaltions, I suggest to use QA benchmarks for experiments rather than captioning benchmark. When things come to LLM, the current captioning metrics such as B@4 and CIDEr might not be ideal to reflect model ability.

[1] Is Space-Time Attention All You Need for Video Understanding?
[2] UniFormer: Unified Transformer for Efficient Spatiotemporal Representation Learning
[3] Dual-AI: Dual-Path Actor Interaction Learning for Group Activity Recognition
[4] MVBench

**Questions:**

1. In Table 1, Q-Former and BLIP-2 are listed as two separate columns. Can you explain this? Q-Former serving as a multi-modal alignment and interaction module, is a part of BLIP-2. It might be doubtable that if the comparison in and conclusion from Table 1 is still plausible.
2. Can you provide formula for Q-Former in Table 1, which takes double FLOPs than the ST Q-Former?
3. Can you provide more details about the computation cost and efficiency, e.g. the number of tokens of LLM input, number of embeddings of Q Former input?
4. What is the specific design of the expert vision transformers you adopted in ST Q-Former? It is okay to adopt directly from existing works, however, implementation details should be provided for reproduction.
5. Is the processing of summary-query identical to SP and T queries, inclusive of the CA operations?
6. For Table 6, what do you mean by SP QFormer? from previous formulation, it should be ST QFormer. If this "average pooling + SP QFormer" including spatial, temporal and summary queries. Please show what the spatial, temporal, summary features are used, as in Figure 3.

---

### Meta-Review · Area_Chair_Catt · 2024-12-15

**Metareview:**

The paper received all negative ratings initially. The authors have not provided the rebuttals. The final recommendation is rejection.

**Additional Comments On Reviewer Discussion:**

N/A.

---

### Decision · Program_Chairs · 2025-01-22

Reject